

# 1 The use of regression for assessing a seasonal forecast model

# 2 experiment

Rasmus E. Benestad[1], Retish Senan[2], Yvan Orsolini[3]
[1]The Norwegian Meteorological Institute, Oslo, 0313, Norway
[2]ECMWF, Reading, U.K.
[3]The Norwegian Institute for Air Research, Kjeller, Norway
*Correspondence to*: R.E. Benestad (rasmus.benestad@met.no)
**Abstract.**We demonstrate how factorial regression can be used to analyse numerical model experiments, testing the effect of
different model settings. We analysed results from a  coupled atmosphere-ocean model to explore how the different choices
in the experimental set-up influence the seasonal predictions. These choices included a representation of the sea-ice and the
choice of top of the atmosphere, and the results suggested that the simulated monthly mean temperatures poleward of the
mid-latitudes are highly sensitivity to the specification of the top of the atmosphere, interpreted as the presence or absence of
a stratosphere. The seasonal forecasts for the mid-to-high latitudes were also sensitive to whether the model set-up included
a dynamic or non-dynamics sea-ice representation, although this effect was less important than the role of the stratosphere.
The temperature in the tropics was insensitive to these choices.
**1 Introduction**
The question of whether seasonal forecasting has useful skill is getting increasingly relevant with the progress in climate
modelling. Another question is how we can learn more about such skills, and one strategy is to examine the models used in
seasonal forecasting. These include state-of-the-art coupled atmosphere-ocean-land-surface models, built on our knowledge
of physical processes and formulated in terms of computer code (Palmer and Anderson, 1994; Stockdale et al., 1998; Palmer,
2004; George and Sutton, 2006). They can be used for seasonal forecasting if a correct initial state is provided, and from
which the subsequent evolution can be simulated. Their skill depends on several factors, such as the quality of the initial
states, the representation of all relevant processes, and whether the seasons ahead truly are predictable in the presence of
non-linear chaos (Palmer, 1996). Thus, in order to address the initial question of useful skill for seasonal predictions, we



need to understand what is important and what is irrelevant for the outcome of the predictions which includes choices about
the model set-up. We know that the atmosphere in the high latitudes is subject to non-linear dynamics, and that the effect of
different factors may interfere and amplify or dampen each other (Charney, 1947; Gill, 1982; Lindzen, 1990; Held, 1993;
Feldstein, 2003).

**1.1 Background**

31          It is well-known that numerical weather prediction (NWP) has a limited forecast horizon because small initial errors

will grow over time in a non-linear fashion (Lorenz, 1963). The case for seasonal forecasting is somewhat different, as it
relies on slow changes in the ocean and cryosphere, which act as persistent boundary conditions. NWP and seasonal
forecasting represent two types of predictability referred to as 'type 1' and 'type 2' (Palmer, 1996). Whereas NWP is more
an initial value problem ('type 1'), the seasonal forecasts embeds a degree of the boundary value problem aspect ('type 2').
Furthermore, seasonal forecasts tend to present the statistics of the weather over a given interval, rather than the exact state at
any instant. In other words, seasonal forecasts, can be compared with predicting a change in the statistics of a sample of
measurements, whereas weather forecasting is more like predicting the details about one specific data point in that sample.

39          Models used for seasonal forecasting have traditionally involved a model for the atmosphere coupled to an ocean

component, and were originally developed for the tropical region and the El Niño Southern Oscillation (Anderson, 1995;
Stockdale, et al., 1998; Palmer and Anderson, 1994). Aspects such as sea-ice, the troposphere and snow cover were not
emphasised as they were not believed to play an important role for the seasonal weather evolution. More recent studies have
looked at the potential influence from sea-ice (Balmaseda et al., 2010; Petoukhov and Semenov, 2010; Overland and Wang,
2010; Francis et al. 2009; Deser et al, 2004; Magnusdottir et al., 2004; Seierstad and Bader, 2008; Benestad, et al. 2010;
Orsolini et al. 2012), especially after the recent dramatic downward trends in the sea ice extent (Kumar et al. 2010; Boé, et
al., 2010; Holland et al., 2008; Wilson, 2009; Kauker et al., 2009; Stroeve et al., 2007, 2008). Other studies have involved
the effect of snow-cover on the atmospheric circulation (Cohen and Entekhabi, 1999; Ge and Gong, 2009; Ueda et al., 2003;
Hawkins et al., 2002; Watanabe and Nitta, 1998; Orsolini et al, 2013) or the influence of stratospheric conditions on the
lower troposphere (Baldwin et al., 2001, 2003; Thompson et al. 2002). Few of these studies, however, have looked at how
these different factors in combination may interfere with each other. Nor has there been many sensitivity tests for
investigating how the model set-up with different combinations of the components representing these different aspects affect
the results. One question we would like to address is whether the response to these different factors add linearly or if the



response is a non-linear function of these factors. Furthermore, it is interesting to find out which of these factors are more
dominant than others. Moreover, our objective was to try to understand which *processes simulated by the model* are more
important, rather than what real signals there are in nature. In this sense, this was a so-called *perfect model study* (Day et al.,
2014). We present the combination of an experimental design (Williams, 1970; Kleijnen and Standridge, 1988) and
analytical techniques that can address this question. The results were taken from a 'synthesis' experiment with a moderately
high-resolution earth system model. Hence, these numerical experiments constitute a kind of sensitivity study (Bürger et al.,

59  2013).


## 2 Method & Data

### 2.1 Model simulations

The model used in this study is the EC-Earth version 2.1 state-of-the-art earth system model (Hazeleger et. al, 2010), which
has been developed by a consortium of meteorological Institutes/Universities across Europe. The atmospheric component of
the EC-Earth model is  based on ECMWF's Integrated Forecasting System (IFS) cycle 31R1 with a new convection scheme
and a new land surface scheme. The ocean component is based on version 2 of the NEMO model (Madec, 2008), with a
horizontal resolution of nominally 1x1 degrees and 42 vertical levels. The sea ice model is the LIM2 model (Fichefet and
Maqueda, 1997). The ocean/ice model is coupled to the atmosphere/land model through the OASIS 3 coupler (Valcke, 2006).
The synthesis experiments consists of a set of 12 coupled model simulations. Six of these simulations used the L62 vertical
resolution for the atmospheric component which extends up to 5 hPa, while the other six used the higher resolution L91
version, which extends up to 0.01 hPa. These two sets of experiments were designed to determine the sensitivity of model
results to a better representation of the stratosphere. Further to evaluate the role of sensitivity to the representation of sea-ice,
the LIM2 sea-ice model was implemented as a standard thermodynamic-dynamic model (DyIce) and as a thermodynamic
only model (NoDyIce). Finally, sensitivity to initial conditions were tested by introducing perturbations to initial conditions
corresponding to positive/negative NAO SST anomaly patterns over the North Atlantic (Melsom, 2010) All simulations
started on 1 Jan 1990 and lasted 90 days. An overview of the model simulations are listed in Table 1.

### 2.2 The analysis



Here the experiments and analysis used an approach known as 'factorial design' (Yates and Mather, 1963; Fisher, 1926; Hill
and Lewicki, 2005; Wilkinson and Rogers, 1973; Benestad et al., 2010), where a factorial regression was used to assess
which influence each of the choices in the model set-up has on the forecasts. Factorial design is a technique that is well-
suited for analysing a set of factors which are considered to have potential effects on the outcome in experiments, where an
analysis of variance (ANOVA; Wilks, 1995) provides estimates for error bars and the level of statistical significance. Hence,
the factorial regression offers an alternative to traditional ways for estimating statistical significance used in meteorology and
climate sciences, such as difference tests between two ensembles. Factorial regression is especially handy when data is
generated by a process which involves two or more factors (set-up options or categories) that are difficult to quantify due to
their discrete nature (e.g. some factors may either present or absent), and has been used to analyse the effect of introducing
different crop varieties in agriculture (e.g. Baril et al. 1995; Vargas et al. 1999; Vargas et al. 2006; Voltas et al. 2005). It is
based on the concept "factorial experiment", or "factorial design", in statistics which involves two or more factors each of
which can be assigned a category or a discrete value. The analysis takes into account all possible combinations of levels over
all such factors including their interactions.
The model response to different initial conditions or different model set-up with different options for three configurations
(SST perturbation, model top, and sea-ice model) was investigated, and a comparison was made between the different
experiments in terms of vertical and horizontal cross sections of temperature anomalies. If the final response ΔT is a linear
function of sea-ice, SST, and stratospheric effects, then it can be expressed as a sum of these different contributions $\Delta T = x_1$
$C(sea\text{-}ice) + x_2 C(SST) + x_3 C(stratosphere)$. The factorial regression provided an estimate of the coefficients $x_i$ and their
error estimates. In a non-linear case, this linear expression was unlikely to provide a good description, and the regression
analysis will yield large errors and low statistical significance.
We do not know the relative strength of the different factors in terms of an input, however, the factorial regression quantifies
the differences between output from different combinations of subsets. It was also used to estimate the probability that the
response in the different combinations of these subsets would be due to chance. The results from the factorial regression
were subsequently used to explore the combined effect of several factors.
The Walker test was used to assess the false discovery rate of the p-values found in the factorial regression (Wilks, 2006).
The test involves comparing the minimum p-value $p_n$ from the local tests with $p_W = 1 - (1 - \alpha)^{1/K}$ for $K$ locations and the
statistical significance level α. If  $p_n < p_W$ then the expected fraction of local null hypothesis with incorrect rejections is
smaller than the number of statistically significant local p-values.



## 3 Results

Figures 1-2 show the difference in the forecasts associated stratosphere, more specifically between the low (L62) and high (L91) top versions of the atmosphere for month 3. Figure 1 shows horizontal transects at 200 and 50 hPa levels respectively. They show the monthly mean temperature starting with a 2-month lead time, and the left panels show results with no initial perturbation (neutral NAO conditions), the middle panels show results from model simulation with initial conditions set at NAO, and the right panels results for which the initial conditions were the negative phase of the NAO. All the panels show that there were differences between the low and high top results, and the difference between the low and high-top model simulation is most pronounced at negative and positive NAO-type initial conditions (not shown). Hence, the forecasted air temperature is sensitive to the inclusion of the upper part of the atmosphere, and the effect can be seen extending throughout the entire vertical extent of the atmosphere (not shown). The difference between the upper and lower rows show the effect of dynamic versus non-dynamic sea-ice representation. With a non-dynamic sea-ice, the inclusion of a stratosphere resulted in stronger vertical dipole patterns at certain longitudes and for positive NAO initial conditions. For the negative NAO initial conditions, the dynamical sea-ice representation enhanced the differences between the L91 and L62 model simulations.

Figure 1 suggests that the effect of including the stratosphere and the representation of sea-ice matter for the mid-latitude to the polar regions, and the choice of the vertical levels had less impact in the tropics. The response suggests mid-latitude wave-like structures in the 200 hPa temperatures, albeit with a tendency of a coherent anomaly over the North Pole. The choice of the sea-ice representation had a pronounced impact on the simulation of the monthly mean temperature after 3 months. The horizontal picture at 50 hPa (Figure 1) suggests radically different wave structure for the negative NAO phase, however, whereas the 'positive' and 'neutral' NAO states differences are more in the details and magnitude. The exact geographical structure in these maps are not the important point here, as the longitude of action will depend on the initial condition. The important information here is the pronounced response in the mid-to-high latitudes.

In summary, it is apparent from Figures 1-2 that the effect of different model aspects such as the choice of model top and sea-ice representation influence the model forecasts. Furthermore, we see that the influence varies with the initial SST conditions, and that different sea-ice representation may introduce changes in the forecast of similar magnitude as the influence of the model top. It is difficult to compare these effects with that of the initial conditions merely from Figures 1-2, however, we can compare the effect from these different aspects through the means of a factorial regression. The analysis of variance for the factorial regression yields a set of coefficients $\beta$ describing the association between the temperature and the model set-up choice, as well as the associated error bars $\varepsilon$ and p-values $p$.



Figure 2 represents the coefficients and the error estimates from the factorial regression. The top panel shows the mean air
temperature for the model forecasts with a model set-up of dynamical sea-ice component, no perturbation in the SST, and 62
vertical levels (low top). Panels b-f show difference in the forecasts due to different choices in the model set-up in terms of
the regression coefficients $\beta$, and panels g-e show error estimates for these coefficients. Regions with large values estimated
for the coefficients and large errors suggest a high sensitivity but also that the response cannot readily be attributed to the
given factor. In other words, the level of both the signal and the noise is high. The magnitude of the error was mainly below
3K except for around 100ºE near the 100hPa level, and generally smaller than the influence of the variable. The results
suggests that the results were sensitive to both the representation of the sea-ice and the inclusion of the stratosphere, as well
as the initial conditions. The analysis also suggests that the magnitude of the effect of the sea-ice representation and the
model top was similar to those of the different SST perturbation near 60ºN. Furthermore, the error estimates associated with
the three factors (SST-perturbation, sea-ice representation and atmosphere top) exhibited similar magnitudes and spatial
structure. A comparison between the different panels in Figure 2 suggests that the different choices for model set-up had
similar magnitude on the predicted outcome for all these factors.
Figure 3 shows the ratio response to error for sea ice (upper), positive NAO SST perturbation (second from the top), negative
NAO SST perturbation (third), and the stratosphere L91 (bottom). Only a small region had a response that was greater in
magnitude than the error estimate for the sea ice, whereas for the SST perturbations and the stratosphere, the regions with
response-to-error ratio has a magnitude greater to unity were more extensive. Note, both large negative and positive values
indicate that the signal is stronger than the noise $|\beta/\varepsilon| > 1$ as $\beta$ may be both positive and negative whereas $\varepsilon$ is positive.
The factorial regression gave highest number of low p-values for the stratosphere (L91), followed by the SST-perturbation
(not shown). For most of the 60ºN vertical transect, the sea-ice representation did not yield a large response compared to the
error term. Furthermore, for a global statistical significance level of $\alpha=0.05$ and K=3840, the threshold value for the Walker
test was $p_W = 1.3 \cdot 10^{-5}$. The minimum p-value for sea ice was 0.01, for SST-perturbation $p_n = 9.2 \cdot 10^{-4}$ and the stratosphere $p_n$
$= 1.6 \cdot 10^{-4}$. In other words, the 12-member experiment was not sufficient to resolve the response in the air temperature
forecast at 60ºN for month 3 to the different set-up options, however, the results suggest that the model top had the greatest
impact on the forecast. The lack of a clear dependency between the sea ice representation and the forecast was also found for
the summer in Benestad et al. (2010), and the obscure links between the factors and the response may be explained by the
presence of strong nonlinear dynamics, where one given factor may result in different forecasts depending on other
influences.



The question of degree of nonlinearity can be addressed by comparing the sum of the influence from the different factors
with simulations with and without a set of factors combined. i.e, we check for the equivalency:
$$DyIce\ pNAO\ L91 - NoDyIce\ nNAO\ L62 = (DyIce - NoDyIce)\ nNAO\ L62 +\quad …….. (1)$$
$$NoDyIce\ (pNAO - nNAO)\ L62 + NoDyIce\ nNAO\ (L91 - L62)$$

Here, the LHS of equation 1 (Figure 4a) shows  the difference between the simulation with high top, dynamic sea ice,
positive NAO perturbation (*DyIce pNAO L91*) and that with low top, non-dynamic sea ice, negative NAO (*NoDyIce nNAO*
*L62*). Figure 4a is compared with sum of the differences from individual factors  (RHS of equation 1, Figure 4b). The
comparison shows that the non-linear model response is mainly confined to the mid- to high-latitudes especially in the
northern Hemisphere (Figure 4c), e.g., along the 60ºN transect presented in Figures 3-5.

## 4 Discussion

The set of sensitivity experiments shows that seasonal forecasts at mid-to-high latitudes are sensitive to a number of factors
concerning the model set-up, and that the choice of subjective and subtle options can have as strong effect on the monthly
mean temperature poleward of the mid-latitudes as the initial conditions. A factorial design experiment allows us to assess
the relative magnitudes of different model height with that of different sea-ice or different SST perturbations. We can also
test the response in the model to see if they are close to being a linear superposition of the different single factors, or if the
model response is highly non-linear. The statistical significance is estimated based on the factorial regression. The
magnitude of the effect of the sea ice, SST-perturbations and the model top height were roughly similar, although the
response to the sea ice was somewhat weaker than the others. The lower ratio of estimate-to-error also reflect the degree of
nonlinearity, and the lower p-values associated with the sea-ice may be due to a greater degree of nonlinearity in the
response to the sea-ice representation. The experiment nevertheless suggested that stratospheric conditions are important for
mid-to-high-latitude seasonal forecasting. This experiment was only carried out for the northern hemisphere winter, and may
change with season. The stratosphere decouples in the summer, and there is a hint of a weaker influence from the model top
in the southern hemisphere where there was summer.
There is previous work where model sensitivity and uncertainty have been assessed (e.g. Rinke et al 2000; Wu, et al. 2005;
Pope and Stratton, 2002; Jacob and Podzun 1997; Knutti et al. 2002; Dethloff et al. 2001), however, most of these



assessments have been carried out for climate simulations as opposed to seasonal forecasts. In seasonal forecasting, the
emphasis has been more on multi-model forecasts and their spread (Weisheimer et al. 2009), rather than the configuration of
single models. However, Jung et al. (2012) discussed the effect of the spatial resolution on seasonal forecast based on an
experimental design with a single model. The use of factorial regression has was also discussed by Rinke et al (2000) in
conjunction with climate simulations, and Benestad et al. (2010) used it in a study of seasonal predictability and the effect of
of boundary conditions associated with sea-ice and initial conditions. This study presented applied factorial regression to a
new set of model configuration options, including the model top, the representation of sea-ice, and initial conditions. In this
case, we emphasised the individual factors rather than their interaction because of the limited sample of model runs.

**5 Conclusions**

The sensitivity tests revealed that seasonal predictability of the temperature at the mid-to-high latitudes was as sensitive to
subjective choices regarding the model set-up as the initial SST conditions. The forecasts for high-latitude regions were in
particular sensitive to the model top, but also the representation of sea ice influenced the outcome. Hence, these results
illustrate the difficulties associated with seasonal forecasting at the higher latitudes and has an effect of the forecast skill. The
tropical temperatures were insensitive to these choices, and the sea-ice representation and the stratosphere do not have a
visible effect on ENSO forecasts.

**6 Acknowledgement**

We are grateful to Wilco Hazeleger and the EC-Earth community for providing a stand-alone version of the EC-Earth model,
and Simona Stefanescu at the ECMWF for all her assistance. This work was carried out under the SPAR-project ("Seasonal
Predictability over the Arctic Region - exploring the role of boundary conditions"; Project 178570, funded by the Norwegian
Research Council and the Meteorological Institute), SPECS (EU Grant Agreement 3038378), and the model simulations
used computational resources at NOTUR- The Norwegian metacenter for computational science. The data used in this
analysis can be obtained by contacting the authors.




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

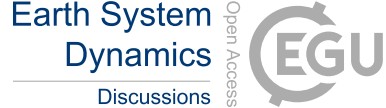


**Figure captions.**
**Figure 1**: Map of monthly mean air temperature difference at 200 hPa between the high-top and low-top experiments for the
third month.
**Figure 2**: Coefficients and error estimates from the factorial regression of air temperature at 60ºN. These results describe the
systematic differences associated between the different choices in the model set-up.
**Figure 3**: The ratio of the factorial regression coefficients to the error estimate for different factors: (a) sea ice representation,
(b) positive NAO SST perturbation, (c) negative NAO SST perturbation  and (d) the model top L91/stratosphere (bottom).
**Figure 4**: Monthly mean air temperaure at 60ºN. (a) Difference between *DyIce pNAO L91* and *NoDyIce nNAO L62* (b) Sum
of the differences: *NoDyIce (pNAO - nNAO) L62*, *(DyIce - NoDyIce) nNAO L62* and *NoDyIce nNAO (L91 - L62)* (c)
Difference (a) - (b).









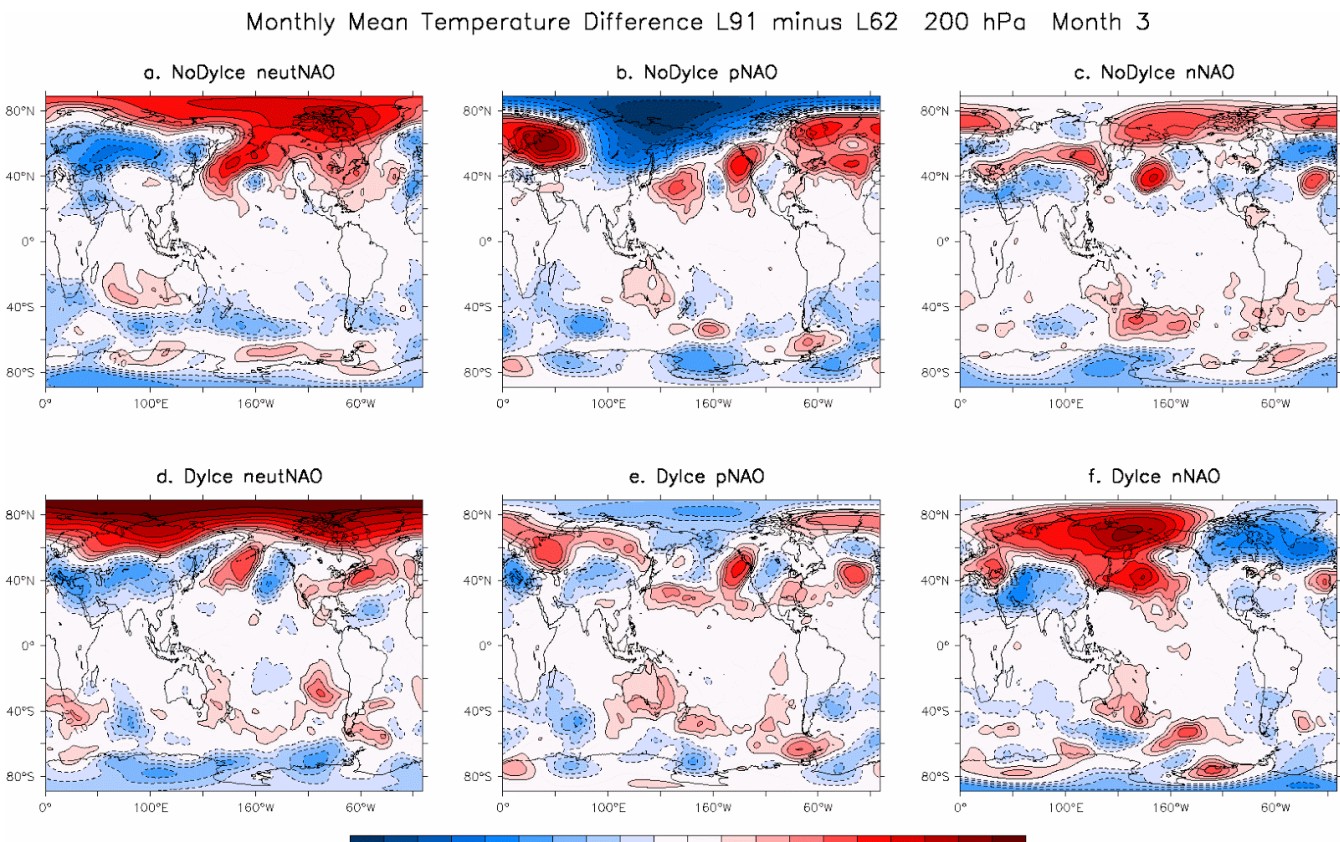

Figure 1






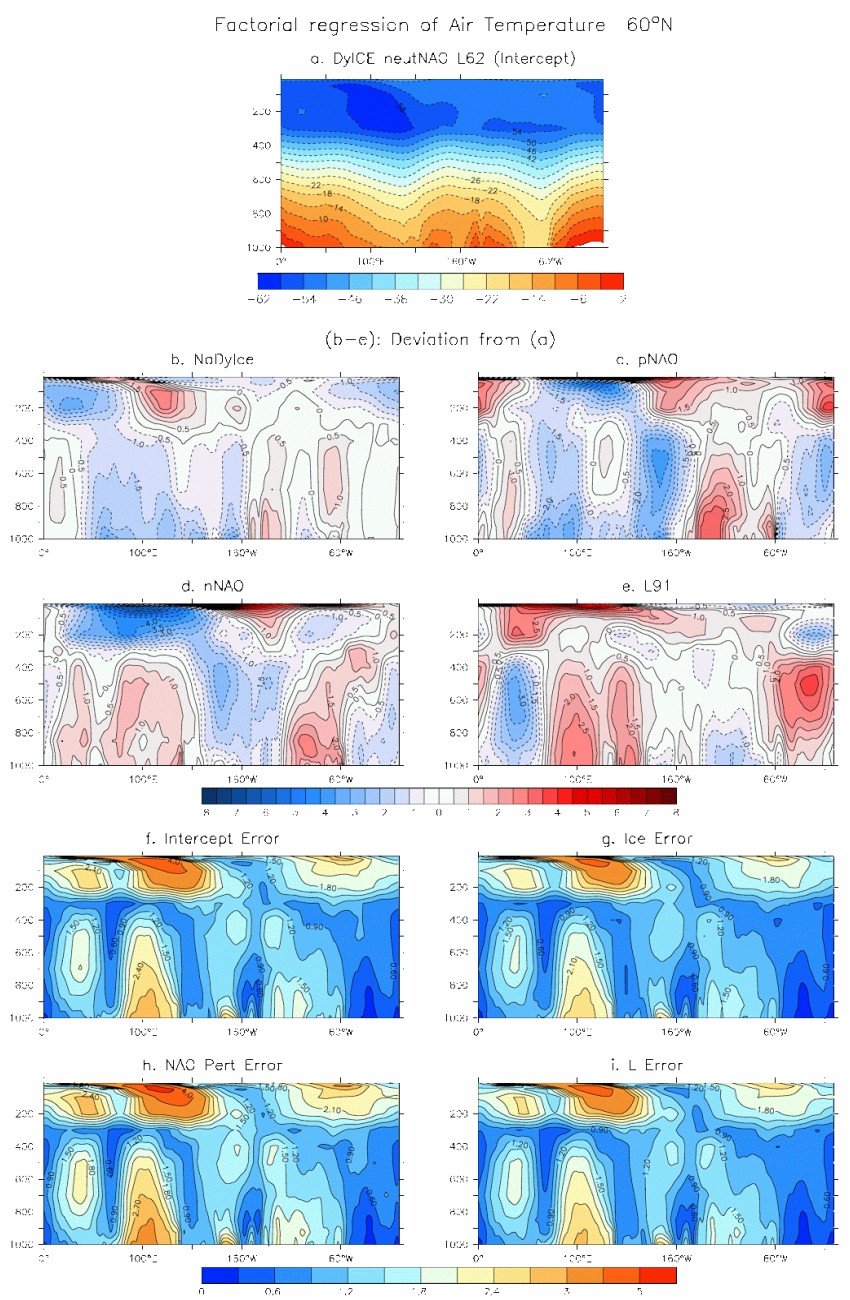

Figure 2





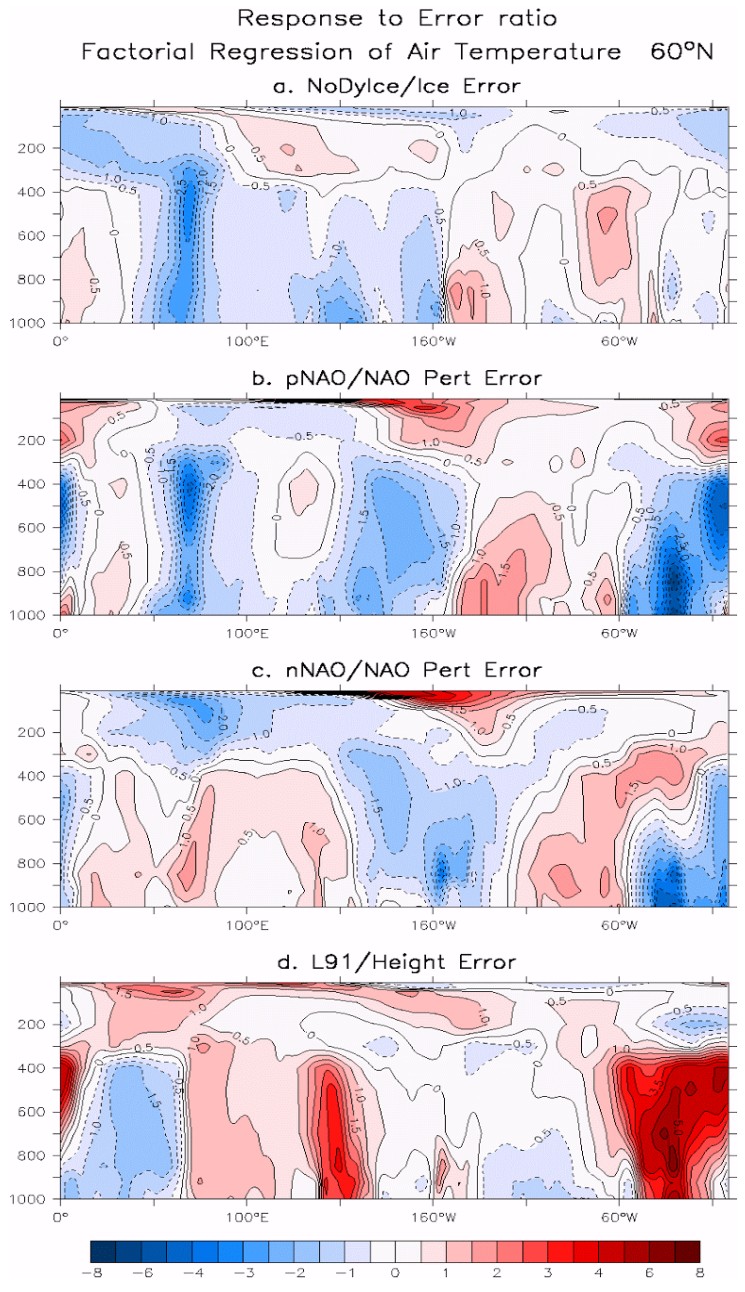

Figure 3





Figure 4

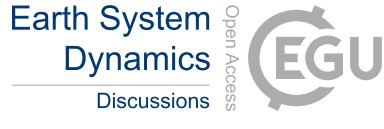



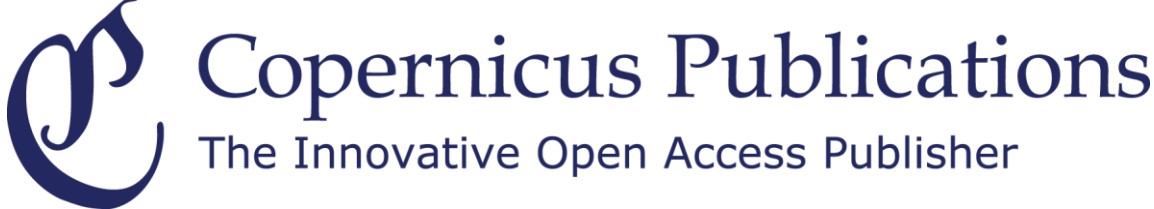


**Figure 1: The logo of Copernicus Publications.**