# Peer review of "The use of regression for assessing a seasonal forecast model experiment"

_Earth System Dynamics, 2016_

## Referee Comment (RC1) · Anonymous Referee #1 · 26 Jul 2016

The authors employ factorial regression to assess the influence of model set-up and initial conditions on seasonal forecasts of the EC-Earth2.1 earth system model; particularly the influence of sea-ice, the stratosphere and NAO SST's initial conditions on monthly mean air temperature anomaly forecasts. The Walker test is employed to assess the global statistical significance of the model response to these three factors. The authors conclude that seasonal predictability of monthly mean temperature in the mid-to-high latitudes is sensitive to the various choices of model set-up and initial SST conditions, with the most sensitive response to the stratosphere and SST initial perturbations, and sea-ice to a lesser degree. Also, the authors argue that the model response in the mid-to-high latitudes is partly due to the non-linear interaction of these three factors.

**Major:**

The problem studied by the authors is relevant and the methodology seems appropriate. However, the results shown are hard to follow at times and the conclusions are not totally convincing. This is partly because of the disconnect between figures, table and explanations, as well as seemingly contradicting results. Specifically:

1. When describing Figure 2, the authors state that "A comparison between the different panels in Figure 2 suggests that the different choices for model set-up had similar magnitude on the predicted outcome for all these factors", which seems to contradict the conclusions of this paper. On the other hand, when describing Figure 3, the authors observe that response-to-error ratio for the different factors are less than unity for sea ice in most regions, and greater than unity in more extensive regions for SST initial perturbations and stratosphere. This is more in tune with the conclusions of the paper. Then, if the response-to-noise ratio of the coefficients is what matters, why showing Figure 2? It is a bit confusing. Figure 3 seems to be showing exactly the ratio of the values shown in the panels of Figure 2. If so, what useful information or conclusion can we draw from Figure 2 alone?

2. The authors obtain minimum p-values for sea ice, SST-perturbation and stratosphere larger than the Walker p-value (**page 6, lines 155–156**). This implies that the Walker test for field significance fails, and so it suggests that the difference between temperature responses for the different model set-ups and SST initial conditions considered here are not statistically significant at the global $\alpha$ level employed. Thus, how robust are the conclusions of this paper? Did the authors consider air temperature forecasts at lead times other than 3 months and/or vertical transects at latitudes higher than 60°N? Did the authors consider more traditional tests than factorial regression for local assessment to check whether they lead to minimum p-values smaller than the Walker p-value? Could the authors expand on this?

3. The authors address the non-linear dependence on sea-ice, SST and strato-spheric effects by comparison of the difference of temperature response of two model configurations with the linearized difference of temperature response, rel-ative to these three factors, of the two model configurations - that is, by com-parison of the left and right hand side of Equation 1 (**Page 7, lines 164–165**). This comparison led the authors to conclude that the non-linear model response is confined mainly to the mid-to-high latitudes where the difference between left and right hand sides of Equation 1 is the largest. Could the interpretation of this result be misleading if other factors are taken into account? For example, could the model response still be approximately linear (i.e., left and right hand sides of Equation 1 approximately equal) if other factors (e.g., snow cover) are considered?

4. According to the authors, factorial regression "offers an alternative to traditional ways for estimating statistical significance", and has not been widely applied to seasonal forecasts. The reader would greatly benefit from a more in depth expo-sition of this methodology in the context of the problem examined by the authors. For example, the authors state that (**page 4, lines 94–95**) "If the final response $\Delta T$ is a linear function of sea-ice, SST, and stratospheric effects, then (...) "

$$\Delta T = x_1 C(sea\text{-}ice) + x_2 C(SST) + x_3 C(stratosphere)$$

What is $C(\cdot)$? What does $C(\cdot)$ represent? Should it be interpreted as a categor-ical predictor depending on its argument? What does this actually mean in this context? Could the authors give a bit more detail about the implementation of this multiple-linear regression model? Should there be a residual added to the right hand side to account for other sources (i.e., factors) of changes in mean temperature?

5. **Page 3, line 76:** The authors state that "An overview of the model simulations are listed in Table 1". I could not find Table 1.

6. **Page 5, lines 107–133:** The authors refer to Figure 1-2 but only describe Figure 1.

7. **Page 5, line 108:** "Figure 1 shows horizontal transects at 200 and 50 hPa levels respectively". Is this correct? It seems that Figure 1 only shows horizontal transects at 200 hPa for different model set-up and SST initial perturbations.

8. **Page 6, line 123:** "The horizontal picture at 50 hPA (Figure 1) suggests ...". Is this shown?

9. **Page 6, line 136:** When referring to Figure 2, it says "Panels b-f show difference in the forecasts due to different choices in the model set-up in terms of the regression coefficients $\beta$, ...". Should it be Panels b-e instead?

10. **Page 6, line 137:** When referring to Figure 2, it says "... panels g-e show error estimates for these coefficients." Should it be panels f-i instead?

11. **Page 6, lines 137–139:** When referring to Figure 2, the authors state "Regions with large values estimated for the coefficients and large errors suggest a high sensitivity ...". Large values relative to what? Are these coefficient and errors given in units of temperature? Is not the response-to-noise ratio what actually matters?

12. **Page 7, line 164–165:** The first term on the right hand side of Equation 1 is "(DyIce-NoDyIce) nNAO L62". Is this correct? I would expect "(DyIce-NoDyIce) pNAO L91" to be consistent with the difference in the left hand side. This also applies to the title of Figure 4b and the caption to Figure 4.

13. **Page 7, line 170–171:** "The comparison shows that the non-linear response ... presented in Figures 3–5". I could not find Figure 5.

[Figure]

14. Labels in figures are generally too small, particularly in Figure 2. In most cases, axes titles and legends in colorbars are missing, e.g., y-axes and colorbars in Figures 1, 2, 3. Also, should the title of Figure 4 be "difference of monthly mean temperature" instead of "monthly mean temperature"?

**Minor:**

The authors should also address the following issues:

1. **Page 1, line 8** Consider replacing "demonstrate". Perhaps, "show"?

2. **Page 5, lines 110–11** At the end of line 110 and beginning of line 11, the word "positive" seems to be missing (i.e., "positive NAO").

3. **Page 7, lines 176–178** "We can also test ... if the model response is highly non-linear". Is this assertion accurate? What if more factors are needed for linear dependence?

4. **Page 8, line 191** Remove "has"

5. **Page 8, line 193** Remove "presented"?

6. **Pages 9–14, lines 214–343** The authors should use a unified style for the references.

---

## Referee Comment (RC2) · Anonymous Referee #2 · 17 Aug 2016

To disentangle the effect of different experimental settings is clearly an interesting point in understanding model dynamics, so the scope of the paper is of great interest. The method of factorial regression itself is not new but this approach has not been extensively applied in the context of long-range forecasting, so it can be considered as an interesting new approach. However, the presentation of methods and results is not easy to read and to follow.

Major points to consider:

The experimental design:

Model simulations: to investigate seasonal winter forecasts, it is at least uncommon to start in January. After its spin up time the model finds itself in the outgoing winter and transition time to spring. The winter jets in the stratosphere and mesosphere are

retreating towards calm conditions during the equinox. Sea-ice might be interesting because it is still growing. However, it would be perhaps easier to investigate pure winter conditions (starting in November and considering December/January/February), when the middle atmosphere is most actively involved.

Initial conditions are unclear, especially with regard to the NAO –experiments. Is the Model started from ERA- interim conditions?

Introduced scientific methods:

Factorial regression: personally, I dislike an introduction of a new method by describing it as "well-suited and handy" at the beginning. This is a conclusion which can be reached at the end and after the reader has had a chance to reenact the features of the introduced new method. The explanation given later in the paper should be moved up and be better explained.

Representation and description of the results:

Table 1 is missing, which makes it difficult to remember the abbreviations of the experimental names. Figure 1 shows results for 200 hPa while in the text a structure at 50 hPa is described which I could find nowhere.

Further, when investigating middle atmosphere dynamics the vertical extension of the figures would look more appropriate if the stratospheric levels can be clearly seen, i.e. a logarithmic vertical axis up to 1hPa.

The results given do not really represent the mid-to-high latitudes, since either just 60°N or 200hPa (why?) as slices were chosen, which makes the statistical investigation for the Northern Hemisphere questionable.

The naming convention is not consistent, making it difficult to follow the discussion: e.g. in case of "response-to-error ratio" or "ratio of estimate-to-error"

The description of Figure 2 seems to mix up the panels
Conclusion of the paper:

Robust conclusions cannot be made because of the limited number of ensemble members given the number of sensitivity experiments.

That the stratosphere is giving a strong response is not surprising because of its dynamical structure and features during the course of the (even retreating) winter allowing the planetary waves to propagate upwards and trigger downward coupling. The role of sea-ice might not change that much from January to March, because the ice-extension is already large and any variability arising from ice-growth would be small.

Minor points:

Not being a native english speaker myself I have the impression that past and present tenses are quite mixed, making the reading also difficult. E.g.:

Lines 95-96 "the factorial regression provided" and "this linear expression was"

General remark

The topic presented in text and figures is not well enough explained to be published now. However, if the statistical investigation made here would be more linked to the already existing knowledge of Northern Hemisphere winter conditions, it would help to accept new statistical methods to be used by the climate modelling community.

―――――――――――――

---

## Short Comment (SC1) · 24 Aug 2016

Sorry. This was posted under wrong reviewer. Reposted.
* * *

---

## Author Comment (AC1) · 24 Aug 2016

We are grateful for the comments raised by reviewer 1, which reveals the need for more careful explanation about the model outcome.

We will revise the paper to make the message and description clearer, with a better connection between figures, tables, and the main text. There may also have been some confusion arising from the question of sensitivity and the identification of a systematic effects associated with different model options, which we will try to explain more carefully in the revised version.

We argue that the sensitivity to various model set-up options are key in terms of predictability, however, the systematic effect can be used for bias correction but may be more difficult to predict. Our model experiment involved an ensemble too small for

answering the latter, but we can deduce a high sensitivity from our runs.

1. The information conveyed through Fig 2 is the raw results of the factorial regression analysis applied to the model experiments - both regression coefficients (intercept and anomalies in terms of this based on different model set-up option) and error estimates. This will be explained more carefully in the revisions. It shows the results of the factorial regression analysis applied to the results from the model experiments. The sensitivity to different sea-ice model options is slightly less than the others, but not by a whole lot. The differences are mainly in the regional anomalies, and the response-to-noise ratio is affected by whether the error estimates are higher in the same region. The small ensemble size used in this experiment precludes high precision when it comes to details.

2. The results are clear - subjective choices about model settings such as choice of atmosphere top (vertical levels) and representation of sea-ice has an effect on the predictions. However, the ensemble we used was too small to detect a robust effect (p-values) in the sense of a systematic bias associated with the settings. In our study of predictability, we limited the test for differences due to model set-up options (this, however, does not apply to different initial conditions) to the final month, which is expected to indicate the largest sensitivity to the choice. Operational seasonal forecasts are usually made for a three-month period, and first and second months are expected to show smaller differences and the effect is not as visible. We also looked at 70N, which doesn't change much, however, it's more interesting to look at a latitude over e.g. Oslo in terms of seasonal predictability. We will expand on this in our revision. The Walker test is applied to traditional assessments such as the the p-value from regression (which is pretty standard), and is not a replacement for each individual test. A chi-squared could also provide a similar base for a Walker test, but was not done as regression analysis was considered to be the best choice and sufficient for these purposes.

3. This is a good comment, and the paper needs to explain more carefully that the op-
tions considered in our experiments were expected to have strongest effects in the high latitude regions. There are other factors too which are expected to affect the tropics, however, the scope of this study was limited to the mid- to high latitudes and the search for causes for poor seasonal predictability. Furthermore, such factors are included in the model simulations, but we did not check their effects by including more experiments with changing their set-up options. The response to different model set-ups is nonlinear, but by considering additional snow-cover, the picture could potentially change: it could give a net response that looked more linear, or it could be that changing sea-ice but not snow (or other model aspects) is not really physically consistent. However, the results still indicate that it is easy to get nonlinear biases in model predictions depending on the model settings.

4. Thanks for asking this: C(.) is the change due to option setting, and is the result from the regression coefficient (one number per grid box). There is no need for adding residuals as the experiments were strictly controlled whereby one factor was changed whereas the other unconsidered factors remained the same.

5. Table 1 was missing (an unfortunate glitch), and will be inserted in the revised manuscript.

6. The reference to Fig 2 was left over from an early version with more figures. The reference will be corrected in the revised manuscript.

7. Only 200hPa - 50hPa is not shown here. The reference to 50hPa was for an additional figure that is not longer shown due to similarities.

8. Now. It should be 200hPa. It will be corrected in the revisions.

9. Yes. Thanks for pointing this out!

10. Yes.

11. Both matter for predictability if one does not know which option is best. High sensitivity (large response) and a robust response both indicate that the option setting

has an effect, but the latter indicates that the effect is more the same for all situations.

12. Yes it's correct as the last term represents the difference between L91 and L62.

13. It has been dropped. The text will be fixed in the revised version.

14. We will try to improve the graphics in the revised version.

Minor: thanks for pointing out. Regarding the nonlinear aspects, all other factors are included although have not been changed here.

---

## Author Comment (AC2) · 24 Aug 2016

We are grateful for the comments raised by reviewer 2, which reveals the need for more careful explanation about the model outcome.

The initial conditions (IC) were based on standard initial conditions for the EC-Earth model that were perturbed as explained in Melsom (2010). The most important aspect of the IC involve the ocean state and SSTs, rather than the atmosphere.

"Well-suited" and "handy" have been dropped in the revised version.

Table 1 went missing during the process of formatting the paper (an unfortunate glitch) but will be added back in the revised version.

The purpose of our paper was to look for reasons why seasonal predictability is so low

at mid- to high latitudes, and the figure was intended to show difference between results from L62 and L91 simulations, which limits the extent that can be shown. We hope that this will trigger further interests and more efforts into improving the understanding of exactly which mechanisms and processes are involved and how they are affected. Our study was limited in terms of resources provided, and we did not have the luxury to do this here. We agree that the strong response of the stratosphere is not surprising, and think it's nice to have some examples showing exactly this in the context of seasonal prediction.

There is not much difference between 70N and 60N, and the choice of 60N was motivated by the latitude of the largest Nordic capitals and the question whether the lack of predictability for these was due to incomplete understanding of the effects of subjective model settings.

We will revise the paper and pay attention to the past and present tenses. Thanks for pointing this out.

Description of Fig 2 will be fixed in the revised version.

The point of this paper is to demonstrate that subjective choices in terms of model setup have an effect on the predictability. A more complete understanding of the dynamics of the wintertime northern hemisphere is outside the scope of this paper.

---

## Author Response (AR1)

**Review of the manuscript**

The manuscript has been reviewed, fixing the problems pointed out by the reviewers (see response to the reviewers). In short, Table 1 is now included and the reference to the figures have been corrected. The revised manuscript also includes some additional explanations and more correct use of past and present tenses.

Rasmus

---

## Author Response (AR3)

**Editor Decision: Publish subject to minor revisions (review by Editor)** (11 Sep 2016) by Ben Kravitz

Comments to the Author:

The manuscript has improved quite a bit during the review process. I have a few additional comments that should be addressed before the manuscript is accepted for publication.

1. You need to be clearer on what you are predicting. Seasonal forecasts of what?

The paper now says explicitly that the predictions are for air tempeerature.

2. I'm unclear as to the insight you're providing. As far as I can tell, you have three points: (A) Factorial regression works. (B) Your predicted variable, whatever it is, depends on some things. (C) Sometimes there are nonlinearities. To me, this isn't a really compelling reason to write a paper. What have we learned that's new?

The paper now explains this more carefully in the discussion: the issue is mainly that various model options affect the outcome for seasonal forecasts in the mid-latitudes, that has not received much attention before because the tropics is not sensitive to this.

3. There are numerous issues with language, including typos and incorrect tense. The authors use past tense quite a bit in (in my opinion) inappropriate places. I am willing to let the authors correct this in their revisions, but if they are unable to do so, I will recommend copyediting by the journal at additional publication cost.

The manuscript has been revised with an eye on the tense. The use of tense depends on general facts about methods (present tense) and what was done (past tense). Hence, the mix. That ought to be ok.

4. I am troubled by the comment on lines 205-206. There are many regions where it's quite clear that nonlinear interactions between the different factors are important. Choosing not to address these is a strange choice to me and has resulted in a weaker paper than what I think it could be.

Thanks – this point was not very clear, but the revised manuscript now explains that this issue is addressed through equation 1 and the combaprison of the terms and their sum. The ensemble was too small to get a good result from the interaction terms in the regression itself.

5. Lines 211-212 mention ENSO forecasts. This has not been discussed anywhere previously in the manuscript. How are you able to conclude this based on your results? Is this now an ENSO paper?

The model gave seasonal forecasts for the whole globe, but we paid attention to the mid-latitudes where the seasonal predictability has been low. It is, however, itnteresting to compare with the tropics and ENSO, since this is the region with the highest skill and the least sensitivity to the model set-up options explored here.